# SVD Provably Denoises Nearest Neighbor Data

**Ravindran Kannan**[1]**, Kijun Shin**[2]**, David Woodruff**[2]
[1]Simons Institute, UC Berkeley, `kannan100@gmail.com`
[2]Carnegie Mellon University, `{kijunshi, dwoodruf}@andrew.cmu.edu`

## Abstract

We study the Nearest Neighbor Search (NNS) problem in a high-dimensional setting where data originates from a low-dimensional subspace and is corrupted by Gaussian noise. Specifically, we consider a semi-random model where $n$ points from an unknown $k$-dimensional subspace of $\mathbb{R}^d$ ($k \ll d$) are perturbed by zero-mean $d$-dimensional Gaussian noise with variance $\sigma^2$ on each coordinate. We assume that the second-nearest neighbor is at least a factor $(1 + \varepsilon)$ farther from the query than the nearest neighbor. We assume we are given only the noisy data and are required to find NN of the uncorrupted data. We prove the following results:

1. For $\sigma \in O(1/k^{1/4})$, we show that simply performing SVD denoises the data; namely, we provably recover accurate NN of uncorrupted data (Theorem 1.1).

2. For $\sigma \gg 1/k^{1/4}$, NN in uncorrupted data is not even **identifiable** from the noisy data in general. This is a matching lower bound on $\sigma$ with the above result, demonstrating the necessity of this threshold for NNS (Lemma 3.1).

3. For $\sigma \gg 1/\sqrt{k}$, the noise magnitude ($\sigma\sqrt{d}$) significantly exceeds the inter-point distances in the unperturbed data. Moreover, NN in noisy data is different from NN in the uncorrupted data in general.

Note that (1) and (3) together imply SVD identifies correct NN in uncorrupted data even in a regime where it is different from NN in noisy data. This was not the case in existing literature (see e.g. (Abdullah et al., 2014)). Another comparison with (Abdullah et al., 2014) is that it requires $\sigma$ to be at least an inverse polynomial in the ambient dimension $d$. The proof of (1) above uses upper bounds on perturbations of singular spaces of matrices as well as concentration and spherical symmetry of Gaussians. We thus give theoretical justification for the performance of spectral methods in practice. We also provide empirical results on real datasets to corroborate our findings.

## 1 Introduction

The nearest neighbor problem is a fundamental task in various fields, including machine learning, data mining, and computer vision. It involves identifying the data point closest to a given query point within a dataset. While conceptually straightforward, the performance and reliability of nearest neighbor search (NNS) can suffer in the presence of noise, particularly in high-dimensional spaces. Real-world data is susceptible to noise, which can ruin the true underlying structure and lead to erroneous nearest neighbor identifications. This necessitates robust techniques that can reduce the impact of noise to ensure accurate and reliable NNS. In this paper, we analyze the NNS problem in a noisy high-dimensional setting. Specifically, we consider a semi-random model where data points from an unknown $k$-dimensional subspace of $\mathbb{R}^d$ ($k \ll d$) are perturbed by adding $d$-dimensional Gaussian noise $N_d(0, \sigma^2 I_d)$ to it.

A fundamental tool in high-dimensional computational geometry, often applied to the NNS problem, is the random projection method. The Johnson-Lindenstrauss Lemma (Johnson and Lindenstrauss, 1984) demonstrates that projecting data onto a uniformly random $k$-dimensional subspace

of $\mathbb{R}^d$ approximately preserves distances between points, offering a computationally efficient way to reduce dimensionality. This approach has had a tremendous impact on algorithmic questions in high-dimensional geometry, leading to the development of algorithms for approximate NNS, such as Locality-Sensitive Hashing (LSH) (Indyk and Motwani, 1998), which are widely used both theoretically and practically. All known variants of LSH for Euclidean space, including (Datar et al., 2004; Andoni and Indyk, 2006; Andoni et al., 2014), involve random projections.

However, it is natural to question whether performance can be improved by replacing "random" projections with "best" or data-aware projections. Practitioners often rely on techniques like Principal Component Analysis (PCA) and its variants for dimension reduction, leading to successful heuristics such as PCA trees (McNames, 2001; Sproull, 1991; Verma et al., 2009), spectral hashing (Weiss et al., 2008), and semantic hashing (Salakhutdinov and Hinton, 2009). These data-adaptive methods frequently outperform algorithms based on oblivious random projections in practice. Yet, unlike random projection methods, these adaptive approaches often lack rigorous correctness or performance guarantees. Bridging this gap between theoretical guarantees and empirical successes for data-aware projections is a significant open question in Massive Data Analysis, see, e.g., (Council, 2013). For worst-case inputs, random projections are known to be theoretically optimal (Alon, 2003; Jayram and Woodruff, 2013), making it challenging to theoretically justify data-aware improvements. This paper aims to provide a theoretical justification for this disparity by studying data-aware projections for the NNS problem.

To address this challenge, we study the semi-random setting proposed in (Abdullah et al., 2014). In this setting, a dataset $P$ of $n$ points in $\mathbb{R}^d$, and a query point $q$ are arbitrarily drawn from an unknown $k$-dimensional subspace (where $k \ll d$) and then perturbed by adding $d$-dimensional Gaussian noise $N_d(0, \sigma^2 I_d)$. The goal is to find the point $p \in P$ that is closest to $q$ in Euclidean distance (considering their unperturbed versions), based on noisy versions.

Our main contribution is a new Singular Value Decomposition (SVD) algorithm for solving the NNS recovery problem. This algorithm can tolerate substantially larger noise levels compared to previous approaches, such as those in (Abdullah et al., 2014). Specifically, we characterize the robustness of NNS under various noise levels. We identify several critical noise level thresholds below in the increasing order of noise level:

- For $\sigma \gg 1/\sqrt{d}$, the noise magnitude (with an expected magnitude of $\sigma\sqrt{d}$) can be substantially larger than the inter-point distances in the original data. Specifically, random Johnson-Lindenstrauss projections will preserve these noisy distances, effectively losing the underlying nearest neighbor structure of the uncorrupted data. Therefore, SVD would be preferred to random projection when $\sigma \gg 1/\sqrt{d}$.

- For $\sigma \in O(1/d^{1/4})$, (Abdullah et al., 2014) proved that the nearest neighbor in the perturbed data remains the same. Their algorithm tolerates a noise level of at most $\sigma = O(1/\sqrt{k}d^{1/4})$, which implies $\sigma$ must be at least an inverse polynomial in the ambient dimension $d$.

- For $\sigma \gg 1/\sqrt{k}$, the nearest neighbor in the perturbed data can, with large probability, differ from the true nearest neighbor.

- For $\sigma \in O(1/k^{1/4})$, our algorithmic results (Theorem 1.1) demonstrate that applying SVD to the perturbed data can effectively identify the true nearest neighbor in this regime. This represents a critical improvement over the previous work of (Abdullah et al., 2014), as our algorithm is effective for $\sigma \gg 1/\sqrt{k}$ where the NN in noisy data is different from the NN in uncorrupted data.

- For $\sigma \gg 1/k^{1/4}$, we show that it is information-theoretically impossible to identify the nearest neighbors from the noisy data. This result complements our algorithmic findings by providing matching lower bounds on the noise level $\sigma$, thereby demonstrating the necessity of the threshold $\sigma = O(1/k^{1/4})$ for NNS.

## 1.1 HIGH-LEVEL OVERVIEW

In addition to improved noise tolerance, our algorithm offers simplicity, requiring only two SVD calls, unlike the iterated PCA approach in (Abdullah et al., 2014). We now discuss the high-level

idea of our algorithm. We represent the input points as the first $n$ columns of a $d \times (n+1)$ matrix $B$, with the last column being the query point $q$. Similarly, we represent the Gaussian noise as a $d \times (n+1)$ matrix $C$ with i.i.d. entries drawn from $N(0, \sigma^2)$. Let $A = B + C$ denote the perturbed data set, which serves as the input to our algorithm. Our approach involves computing the SVD of $A$ and projecting $A$ onto its top $k$-dimensional subspace. A direct application of the SVD was not explored in earlier works to handle such high noise levels. The only earlier work we are aware of with related provable guarantees in a noisy model via the SVD is that on latent semantic indexing (Papadimitriou et al., 2000), though (Papadimitriou et al., 2000) makes strong assumptions.

More specifically, we process the $j$ indices in two parts: first for $1 \leq j \leq \frac{n}{2}$, then for $\frac{n}{2} + 1 \leq j \leq n$. Let $A^{(1)}$ be a $d \times (\frac{n}{2} + 1)$ matrix consisting of the first $n/2$ columns of $A$ and the query point (as column $n/2 + 1$). Similarly, let $A^{(2)}$ be a $d \times (\frac{n}{2} + 1)$ matrix formed by the last $n/2$ columns of $A$ and the query point. The query point is in both parts. This superscript notation, $(1)$ and $(2)$, is also extended to $B$ and $C$. Let $U^{(1)}$ be the subspace spanned by the $k$ top singular vectors of the first $n/2$ columns of $A^{(1)}$ (i.e., $A^{(1)}[1, \frac{n}{2}]$). Similarly, $U^{(2)}$ is the subspace spanned by the $k$ top singular vectors of the first $n/2$ columns of $A^{(2)}$ (i.e., $A^{(2)}[1, \frac{n}{2}]$). Since $A^{(1)}$ and $A^{(2)}$ are given, $U^{(1)}$ and $U^{(2)}$ can be computed. The point of splitting the data into 2 parts is that $P_{U^{(2)}}$ and $A^{(1)}$ are stochastically independent and this makes our probabilistic arguments simpler. It is not clear that this is necassary and we leave it as an open question as to whether the simpler algorithm without splitting provably works.

We denote the projection matrix onto a subspace $U \subseteq \mathbb{R}^d$ as $P_U$. The underlying idea is that projecting points (both data and query) onto the SVD subspace effectively extracts the latent subspace structure, which is sufficient to estimate distances, $||p_i - q||$. Thus, the main algorithm proceeds as follows: to estimate all distances for the first $n/2$ points, we compute the minimum value of:

$$\min_{1 \leq j \leq \frac{n}{2}} \left|\left| P_{U^{(2)}} \left( A^{(1)}_{\cdot,j} - A^{(1)}_{\cdot,n/2+1} \right) \right|\right|,$$

where $A^{(1)}_{\cdot,j}$ denotes the $j$-th column of $A^{(1)}$. With $x_j = \mathbf{e}_j - \mathbf{e}_{n/2+1}$, this expression simplifies to $||P_{U^{(2)}} A^{(1)} x_j||$. Our claim is that, under a specific noise regime, $||P_{U^{(2)}} A^{(1)} x_j||$ provides a $(1 + \varepsilon)$-approximation of $||B^{(1)} x_j|| = ||p_i - q||$ for any $\varepsilon > 0$. Subsequently, similar steps are performed for the second part of the data. The complete algorithm is then as follows:

---

**Algorithm 1** $(1 + \varepsilon)$-approximate NNS for the Semi-Random Model

---

**Require:** An ambient space $\mathbb{R}^d$ and a matrix $A \in \mathbb{R}^{d \times (n+1)}$ representing the perturbed point set.
**Ensure:** Returns the index of a $(1 + \varepsilon)$-approximate nearest neighbor for the unperturbed data.
1: $A^{(1)} \leftarrow$ matrix formed by columns 1 to $n/2$ of $A$ and column $n+1$ of $A$.
2: $A^{(2)} \leftarrow$ matrix formed by columns $n/2 + 1$ to $n$ of $A$ and column $n+1$ of $A$.
3: $U^{(1)} \leftarrow$ the subspace spanned by the $k$ top singular vectors of $A^{(1)}$.
4: $U^{(2)} \leftarrow$ the subspace spanned by the $k$ top singular vectors of $A^{(2)}$.
5: $j_1 \leftarrow \arg\min_{1 \leq j \leq \frac{n}{2}} ||P_{U^{(2)}} A^{(1)} x_j||$.
6: $j_2 \leftarrow \arg\min_{1 \leq j \leq \frac{n}{2}} ||P_{U^{(1)}} A^{(2)} x_j||$.
7: **if** $||P_{U^{(2)}} A^{(1)} x_{j_1}|| < ||P_{U^{(1)}} A^{(2)} x_{j_2}||$ **then**
8:     Return $j_1$.
9: **else**
10:    Return $j_2 + n/2$.
11: **end if**

---

Below, we formalize the theoretical guarantee of Algorithm 1. Let $s_k(X)$ denote the $k$-th singular value of matrix $X$. If $\text{rank}(X) < k$, then $s_k(X)$ is defined as 0.

**Theorem 1.1.** *For the semi-random model described above, if the noise level $\sigma$ satisfies:*

$$\sigma \leq \min \left( \sqrt{\frac{\varepsilon}{240}} \frac{\min \left( ||B^{(1)} x_j||, ||B^{(2)} x_j|| \right)}{(k \ln n)^{1/4}}, \frac{\varepsilon \cdot \min(s_k(B^{(1)}), s_k(B^{(2)}))}{75\sqrt{n}}, \frac{\varepsilon \cdot \min \left( ||B^{(1)} x_j||, ||B^{(2)} x_j|| \right)}{36\sqrt{\ln n}} \right),$$

*Algorithm 1 returns a $(1 + \varepsilon)$-approximate nearest point for $\varepsilon > 0$ with probability at least $1 - \frac{1}{n}$.*

**Remark 1.2** (Interpretation of Bounds). *Theorem 1.1 highlights two distinct scaling requirements for recovery:*

*1. **Intrinsic Dimension** (k): The term $O(1/k^{1/4})$ reflects the geometric complexity of the subspace. This threshold aligns with the information-theoretic limits of preserving nearest neighbor structures (see Lemma 3.1).*

*2. **Signal Strength** ($s_k(B)$): The term proportional to $s_k(B)/\sqrt{n}$ bounds the spectral gap. By Wedin's Theorem, this ratio ensures that the principal angles between the true subspace $V$ and the empirical subspace $U^{(2)}$ remain small. If $s_k(B)$ were too small, the signal directions would be indistinguishable from noise directions, making subspace recovery impossible regardless of the algorithm used. We discuss this factor in Section 2.3.*

This highlights the power of SVD in extracting low-dimensional structure from noisy high-dimensional observations. We also explored its impact through our empirical results. Our empirical results further validate our theoretical findings, demonstrating the practical benefits of our SVD-based approach and its superior performance compared to naïve algorithms, particularly in terms of noise dependence on the intrinsic subspace dimension $k$ and sensitivity to the $k$-th minimum singular value of the data $s_k(B)$.

**Organization:** Section 2 details our algorithmic approach, including the problem setup and the SVD-based algorithm, along with its analysis and discussion. Section 3 provides theoretical lower bounds, demonstrating the optimality of our proposed noise thresholds. Section 4 presents empirical results that validate our theoretical findings and illustrate the practical benefits of our approach.

## 2 ALGORITHMIC RESULTS

### 2.1 THE MODEL AND PROBLEM

We employ a semi-random data model that assumes the original data consists of $n$ arbitrary (not random) points from a $k$-dimensional subspace $V$ of $\mathbb{R}^d$. We also assume the query point lie in $V$. The original data is latent (hidden), and so is $V$. The input is noisy data, obtained by adding Gaussian noise to the original data. Such a semi-random model has been widely used (Abdullah et al., 2014; Azar et al., 2001).

$B$ is a $d \times (n+1)$ matrix where the first $n$ columns represent the latent data points, and the last column represents the latent query. $C$ is a $d \times (n+1)$ matrix representing the perturbations to the $n$ latent data points and the query. We assume the entries of $C$ are i.i.d. random variables, each drawn from $N(0, \sigma^2)$. The observed data $A = B + C$ constitutes the input to the problem. For notational convenience, let $x'_j = \mathbf{e}_j - \mathbf{e}_{n+1}$ such that $B_{\cdot,j} - B_{\cdot,n+1} = Bx'_j$. The objective is to output a $(1 + \varepsilon)$-approximate nearest neighbor for $\varepsilon > 0$. Specifically, the goal is to find an index $j \in \{1, 2, \ldots, n\}$ satisfying $||Bx'_j|| \leq (1 + \varepsilon) \cdot \min_{1 \leq i \leq n} ||Bx'_i||$.

### 2.2 ANALYSIS

We are now ready to start the proof of Theorem 1.1. We said that $\left\|P_{U^{(2)}}A^{(1)}x_j\right\|$ is a good approximation to $\left\|B^{(1)}x_j\right\|$. Below, Lemma 2.1 quantifies how well the projected noisy distances approximate the true latent distances, plus an expected noise term $(2k\sigma^2)$. Hence, we can infer that if $j$ satisfies: $||P_{U^{(2)}}A^{(1)}x_j|| = \min_{1 \leq i \leq n/2} ||P_{U^{(2)}}A^{(1)}x_i||$, then, $j$ approximately minimizes $||B^{(1)}x_j||$ over $||B^{(1)}x_i||$ for $1 \leq i \leq n/2$ within error at most the right hand side of (1).

**Lemma 2.1.** *Assume $B^{(1)}$ has rank $k$, $n \geq d$, and $k \geq \ln n$. Then, for each $1 \leq j \leq n/2$, the following holds with at least $1 - \frac{1}{n^2}$ probability:*

$$\left| \left\|P_{U^{(2)}}A^{(1)}x_j\right\|^2 - 2k\sigma^2 - \left\|B^{(1)}x_j\right\|^2 \right|$$

$$\leq \frac{100\sigma^2 n}{s_k^2(B^{(2)})} \left\|B^{(1)}x_j\right\|^2 + \frac{40\sigma\sqrt{n}}{s_k(B^{(2)})} \left\|B^{(1)}x_j\right\|^2 + 20\sigma \left\|B^{(1)}x_j\right\| \sqrt{\ln n} + 40\sigma^2\sqrt{k\ln n}. \quad (1)$$

*Similarly, assuming $B^{(2)}$ has rank $k$, $n \geq d$, and $k \geq \ln n$, then for each $1 \leq j \leq n/2$, the following holds with at least $1 - \frac{1}{n^2}$ probability:*

$$\left| \left\| P_{U^{(1)}} A^{(2)} x_j \right\|^2 - 2k\sigma^2 - \left\| B^{(2)} x_j \right\|^2 \right|$$

$$\leq \frac{100\sigma^2 n}{s_k^2(B^{(1)})} \left\| B^{(2)} x_j \right\|^2 + \frac{40\sigma\sqrt{n}}{s_k(B^{(1)})} \left\| B^{(2)} x_j \right\|^2 + 20\sigma \left\| B^{(2)} x_j \right\| \sqrt{\ln n} + 40\sigma^2 \sqrt{k \ln n}.$$

*Proof.* Without loss of generality, we prove only the first part. Since all data points and the query point $q$ lie in $V$, projecting $B^{(1)}$ onto $V$ does not change it. Thus, $P_V B^{(1)} = B^{(1)}$. Therefore, the following holds:

$$P_{U^{(2)}} A^{(1)} = P_{U^{(2)}} (B^{(1)} + C^{(1)}) = B^{(1)} + (P_{U^{(2)}} - P_V) B^{(1)} + P_{U^{(2)}} C^{(1)}.$$

We aim to bound each term in this expression. First, $\|P_{U^{(2)}} - P_V\|$ can be bounded in terms of the $k$-th singular value $s_k(B^{(1)})$. Second, $P_{U^{(2)}} C^{(1)} x_j$ is a random Gaussian vector, as per the definition of the noise matrix $C$. Thus, in the lemma statement, terms containing $s_k(B^{(1)})$ relate to the effect of the $(P_{U^{(2)}} - P_V) B^{(1)} x_j$ term, while the remaining terms are associated with the inner product of random Gaussian noise or the norm of the noise vector itself.

Since $P_{U^{(2)}}$ is symmetric, we get:

$$\left\| P_{U^{(2)}} A^{(1)} x_j \right\|^2 = \left\| B^{(1)} x_j + (P_{U^{(2)}} - P_V) B^{(1)} x_j + P_{U^{(2)}} C^{(1)} x_j \right\|^2$$

$$= \left\| B^{(1)} x_j \right\|^2 + \left\| (P_{U^{(2)}} - P_V) B^{(1)} x_j \right\|^2 + \left\| P_{U^{(2)}} C^{(1)} x_j \right\|^2$$

$$+ 2x_j^T B^{(1)^T} (P_{U^{(2)}} - P_V) B^{(1)} x_j + 2x_j^T B^{(1)^T} P_{U^{(2)}} C^{(1)} x_j + 2x_j^T C^{(1)T} P_{U^{(2)}} (P_{U^{(2)}} - P_V) B^{(1)} x_j.$$

$$(2)$$

Now, $P_{U^{(2)}}$ is idempotent: $P_{U^{(2)}} P_{U^{(2)}} = P_{U^{(2)}}$. Also since the columns of $B^{(1)}$ lie in $V$, we have $P_V B^{(1)} = B^{(1)}$. Plugging these into the last term on the right hand side of (2), we see that term is zero $(P_{U^{(2)}} (P_{U^{(2)}} - P_V B^{(1)}) = 0)$. It turns out that each of the other terms can be bounded. By Lemma 2.4, Lemma 2.5, Lemma 2.6, and Lemma 2.7 below, together with the union bound, the theorem is proved. $\square$

Before proving the lemmas directly, we first show a bound on the spectral norm of the matrix $P_{U^{(2)}} - P_V$. Recall that $V$ is the true underlying subspace containing the points and the query, while $U^{(2)}$ is the subspace spanned by the columns of the perturbed matrix $A$. Thus, bounds on the spectral norm of the difference between these two projection matrices can be expressed in terms of the noise $\sigma$ as follows. For this, we use well-established results from Numerical Analysis, namely, the $\sin \Theta$ theorem by (Davis and Kahan, 1970) and the corresponding theorem for singular subspaces due to (Wedin, 1972), which is stated as Lemma 2.2 below:

**Lemma 2.2** ((O'Rourke et al., 2018, Theorem 19)). *Let $B$ be a real $d \times n$ matrix with singular values $s_1 \geq \ldots \geq s_{\min(d,n)} \geq 0$ and corresponding singular vectors $v_1, \ldots, v_{\min(d,n)}$. Also, let $E$ be an $d \times n$ perturbation matrix. Let $s'_1 \geq \ldots \geq s'_{\min(d,n)} \geq 0$ denote the singular values of $B + E$ with corresponding singular vectors $v'_1, \ldots, v'_{\min(d,n)}$. Suppose the rank of $B$ is $r$. For $1 \leq j \leq r$, let $V_j$ and $V'_j$ be the subspaces spanned by $\{v_1, \ldots, v_j\}$ and $\{v'_1, \ldots, v'_j\}$. Then, if $V_j$ and $V'_j$ are both $j$ dimensional spaces, the following holds for the (principal) angle between two subspaces:*

$$\sin \angle (V_j, V'_j) := \max_{v \in V_j, v \neq 0} \min_{v' \in V'_j, v' \neq 0} \sin \angle (v, v') = \left\| P_{V_j} - P_{V'_j} \right\| \leq \frac{2 \|E\|}{s_j - s_{j+1}}$$

*where $s_{r+1} = 0$.*

The bound in Lemma 2.2 is in terms of the $\|E\|$ term, which is the spectral norm of the perturbation matrix. To use this, we need an upper bound on $\|E\|$. For this, we use a well-known result from Random Matrix Theory:

**Lemma 2.3** ((Rudelson and Vershynin, 2010, Equation 2.3)). *Suppose all entries of a $d \times n$ matrix $E$ are sampled from $N(0, \sigma^2)$ i.i.d. Then the following holds for any $t \geq 0$:*

$$\Pr\left[\|E\| > \sigma(\sqrt{n} + \sqrt{d}) + t\right] \leq 2\exp\left(-\frac{t^2}{2\sigma^2}\right).$$

**Lemma 2.4.** *Assume that $n \geq d$, and $B^{(2)}$ has rank $k$. Then for all $1 \leq j \leq n/2$, the following holds with at least $1 - \frac{1}{4n^2}$ probability:*

$$\left\|(P_{U^{(2)}} - P_V)B^{(1)}x_j\right\|^2 \leq \frac{100\sigma^2 n}{s_k^2(B^{(2)})}\left\|B^{(1)}x_j\right\|^2.$$

**Lemma 2.5.** *Assume $k \geq \ln n$ and $n \geq d$. Then $\left|\left\|P_{U^{(2)}}C^{(1)}x_j\right\|^2 - 2k\sigma^2\right| \leq 40\sqrt{\ln n}\sqrt{k}\sigma^2$ holds for all $1 \leq j \leq n/2$, with at least $1 - \frac{1}{2n^2}$ probability.*

**Lemma 2.6.** *Assume $n \geq d$, and $B^{(2)}$ has rank $k$. Then, for all $1 \leq j \leq n/2$, the following holds with at least $1 - \frac{1}{2n^2}$ probability.:*

$$\left|x_j^T {B^{(1)}}^T (P_{U^{(2)}} - P_V)B^{(1)}x_j\right| \leq \frac{4\sigma\sqrt{n}}{s_k(B^{(2)})}\left\|B^{(1)}x_j\right\|^2.$$

**Lemma 2.7.** *For all $1 \leq j \leq n/2$, the following holds:*

$$\left|x_j^T {B^{(1)}}^T P_{U^{(2)}} Cx_j\right| \leq 3\sigma\left\|B^{(1)}x_j\right\|\sqrt{\ln n}$$

*with at least $1 - \frac{1}{4n^2}$ probability.*

We now use Lemma 2.1 to prove the following corollary, which quantifies the noise level tolerance needed for a $(1 \pm O(\varepsilon))$-approximation of the distance:

**Corollary 2.8.** *Suppose the noise $\sigma$ satisfies:*

$$\sigma \leq \min\left(\sqrt{\frac{\varepsilon}{240}}\frac{\left\|B^{(1)}x_j\right\|}{(k\ln n)^{1/4}}, \frac{\varepsilon s_k(B^{(1)})}{75\sqrt{n}}, \frac{\varepsilon\left\|B^{(1)}x_j\right\|}{36\sqrt{\ln n}}\right).$$

*Then, we have*

$$\left\|P_{U^{(2)}}A^{(1)}x_j\right\|^2 - 2k\sigma^2 \in \left[\left(1 - \frac{\varepsilon}{3}\right)\left\|B^{(1)}x_j\right\|^2, \left(1 + \frac{\varepsilon}{3}\right)\left\|B^{(1)}x_j\right\|^2\right]$$

*for all $1 \leq j \leq \frac{n}{2}$ with at least $1 - \frac{1}{2n}$ probability. Similarly, the above holds for $A^{(2)}$ and $B^{(2)}$.*

We now provide the proof of Theorem 1.1, which offers the theoretical guarantee for our main algorithm:

*Proof.* Our algorithm returns the index $j$ corresponding to the minimum value found across two parts of the minimization: $\min_{1 \leq j \leq n/2}\left\|P_{U^{(2)}}A^{(1)}x_j\right\|$ and $\min_{n/2+1 \leq j \leq n}\left\|P_{U^{(1)}}A^{(2)}x_{j-n/2}\right\|$. Without loss of generality, assume that the first column of $B$ (corresponding to $p_1$) is the nearest neighbor to the query point $q$, then we need to show that the algorithm selects an index $j^*$ such that $\left\|B^{(1)}x_{j^*}\right\| \leq (1 + \varepsilon)\left\|B^{(1)}x_1\right\|$.

Suppose our algorithm returns an index $j^*$ from the first part of the minimization, where $1 \leq j^* \leq n/2$. By the algorithm's logic, this implies:

$$\left\|P_{U^{(2)}}A^{(1)}x_{j^*}\right\|^2 - 2k\sigma^2 \leq \left\|P_{U^{(2)}}A^{(1)}x_1\right\|^2 - 2k\sigma^2.$$

Since the noise level $\sigma$ satisfies the conditions of Corollary 2.8, we have:

$$\left(1 - \frac{\varepsilon}{3}\right)\left\|B^{(1)}x_{j^*}\right\|^2 \leq \left\|P_{U^{(2)}}A^{(1)}x_{j^*}\right\|^2 \leq \left\|P_{U^{(2)}}A^{(1)}x_1\right\|^2 \leq \left(1 + \frac{\varepsilon}{3}\right)\left\|B^{(1)}x_1\right\|^2$$

This implies

$$\left\|B^{(1)}x_{j^*}\right\|^2 \leq \frac{1 + \varepsilon/3}{1 - \varepsilon/3}\left\|B^{(1)}x_1\right\|^2 \leq (1 + \varepsilon)\left\|B^{(1)}x_1\right\|^2.$$

Thus, the desired result is obtained. If our algorithm returns an index $j^*$ from the second part of the minimization, a similar argument establishes the correctness of our algorithm. $\square$

## 2.3 DISCUSSION

Our work shows recovery even when the noisy nearest neighbor has changed, whereas (Abdullah et al., 2014) operates in a regime where the NN is preserved despite the noise, which is a key conceptual distinction. However, Theorem 1.1 shows that our algorithm can also be affected by the spectral property of the data matrix. We discuss several aspects of this difference in comparison to the prior work, as well as other aspects of our algorithm below.

$s_k(B)$ **term in Theorem 1.1.** Our noise bound for $\sigma$ depends not only on $O(k^{-1/4})$ but also on the term $s_k(B)/\sqrt{n}$. One might wonder if this term diminishes as the number of data points $n$ increases, thereby weakening our central claim that $\sigma = O(k^{-1/4})$. However, $s_k(B)$ is a property of the entire $d \times (n+1)$ data matrix $B$. As $n$ increases, the matrix itself grows, and its singular values are generally expected to grow as well.

More precisely, for a data matrix $B$ whose columns (the data points) have a reasonably constant average norm, the squared Frobenius norm $||B||_F^2 = \sum_{i,j} B_{i,j}^2$ will grow approximately linearly with $n$. Since the squared Frobenius norm for our rank-$k$ matrix $B$ is also equal to the sum of its squared singular values, i.e., $||B||_F^2 = \sum_{i=1}^k s_i(B)^2$, this growth must be distributed among the singular values. Furthermore, if the data matrix $B$ is *well-conditioned*—meaning its singular values are not pathologically distributed and the data points do not collapse onto a lower-dimensional subspace—then individual singular values are expected to scale proportionally to $||B||_F = O(\sqrt{n})$. Therefore, for well-conditioned data, the ratio $s_k(B)/\sqrt{n}$ is expected to converge to a non-zero constant rather than decay to 0. An example of a well-conditioned matrix family is a random matrix where each entry is sampled independently and identically from a random variable with mean 0, variance 1, and finite fourth moment (Bai and Yin, 1993). They proved that for such an $m \times n$ matrix ($m \leq n$), the smallest singular value is almost surely $(1 - \sqrt{m/n})^2$. The assumption that the data matrix is well-conditioned is standard for many real-world, high-dimensional datasets.

**Comparison to Prior Work (Abdullah et al., 2014).** Prior work, unlike our algorithm, does not require any assumption on the singular values of the (latent) data matrix. However, this generality comes at the cost of a much stricter noise requirement, where the noise level $\sigma$ must be bounded by an inverse polynomial in the ambient dimension $d$. By contrast, our approach introduces a dependency on the data's spectrum $s_k(B)$ but achieves significantly improved noise tolerance with respect to the intrinsic dimension $k$. Our contribution is therefore most impactful in the common scenario where the ambient dimension $d$ is very high, but the data lies near a low-dimensional, well-conditioned subspace (i.e., large $d$, small $k$). We believe this represents a valuable and practical trade-off.

**Connection between Johnson-Lindenstrauss Lemma and Our Results.** The upper bounds of $\sigma$ in Theorem 1 depend on three terms. For the first term to be the smallest, our theorem requires $k = \Omega(\ln n/\varepsilon^2)$. This term $\Omega(\ln n/\varepsilon^2)$ also appears in the standard Johnson-Lindenstrauss (JL) Lemma. The JL Lemma states that a random projection of $n$ data points from a $d$-dimensional ambient space into a $k$-dimensional subspace preserves all pairwise distances up to a $(1 + \varepsilon)$ multiplicative factor. The JL Lemma thus establishes the required dimensionality for an oblivious random projection to preserve the geometry of $n$ points, where $k = \Omega(\ln n/\varepsilon^2)$ is known to be the information-theoretic complexity for the pairwise distance preservation problem.

Our result implies that for SVD to be an effective denoising strategy for NNS, the underlying subspace containing the true signal must itself have a complexity that scales in a JL-like manner. If $k$ were smaller than this threshold, the $k$-dimensional subspace would be too *simple* to robustly encode the identity of the nearest neighbor against noise across all $n$ points. In summary, the JL Lemma focuses on dimensionality reduction (from $d \to k$) using random projections. Conversely, our work addresses denoising when the data already possesses a low intrinsic dimension $k$. We demonstrate that if the data has this structure and $k$ satisfies this fundamental complexity requirement, then using SVD enables accurate NNS recovery in a high-noise regime ($\sigma = O(k^{-1/4})$), a regime where standard JL would fail to preserve the nearest neighbor identity.

**Matrix Split Ratio Choice.** In our algorithm, we split the perturbed point set into two halves. This 50-50 split ratio is actually a minimax optimal choice. The primary goal of splitting the data is to obtain a set of learned singular vectors (e.g., spanning subspace $U^{(2)}$) that are stochastically independent of the noise in the data we intend to denoise (e.g., $A^{(1)}$). The accuracy of this learned subspace $U^{(2)}$ as an estimate for the true latent subspace $V$ depends directly on the number of data points used to compute it (i.e. $(1-p)n$). A smaller partition size leads to a less stable SVD and a larger error in the estimated subspace, as quantified by the bounds on $||P_{U^{(2)}} - P_V||$ in our proofs (Lemma 2.4). Note that the algorithm's overall performance is limited by the weaker of the two subspace estimations. Therefore, due to the inherent symmetry of our algorithm, the $50 - 50$ split maximizes the size of the smaller partition, thereby providing the most robust performance when no prior information about the data's structure is known.

**Runtime of the Algorithm 1.** The running time of our methods just involves two SVD calculations on the two halves of the matrix, and using iterative methods, such as those in (Musco and Musco, 2015), each can be done in $O(ndk/\min(1, \sqrt{\text{gap}}))$ time, up to logarithmic factors, where gap $= s_k/s_{k+1} - 1$. For our recovery guarantees (Theorem 1.1), we require $s_k(B)$ to be sufficiently larger than $\sigma$. Then we expect that $s_k(A) \sim s_k(B)$ and $s_{k+1}(A) \sim \sigma$, which implies gap $= s_k(A)/s_{k+1}(A) - 1$ will be large. This helps both for recovery and for the efficiency of the SVD calculation. We then need to project the data onto the top $k$ components we find, which is $O(ndk)$ time.

**Requirement to Know the Intrinsic Dimension $k$.** In our setting, we need to know $k$ to run our algorithm. In other words, choosing a cutoff $k' > k$ would imply $s_{k'} = 0$, yielding a vacuous bound. However, real-world data is often full-rank with a decaying spectrum. In practice, selecting $k' > k$ is a valid heuristic to capture signal "leakage" into lower singular values without capturing the bulk of the noise spectrum.

**About $K$-nearest neighbors.** Corollary 2.8 yields an estimate of the latent distances after subtracting the constant noise term. Define the estimated squared distance

$$\widehat{d}_j^2 := \begin{cases} \|P_{U^{(2)}} A^{(1)} x_j\|^2 - 2k\sigma^2, & 1 \le j \le n/2, \\ \|P_{U^{(1)}} A^{(2)} x_{j-n/2}\|^2 - 2k\sigma^2, & n/2 + 1 \le j \le n, \end{cases}$$

and let $d_j^2 := \|Bx_j'\|^2 = \|p_j - q\|^2$ denote the corresponding latent distance (up to the half-splitting reindexing). Under the same noise regime as Corollary 2.8, with high probability we have, simultaneously for all $j \in [n]$,

$$\widehat{d}_j^2 \in \left[ \left(1 - \tfrac{\varepsilon}{3}\right)d_j^2, \ \left(1 + \tfrac{\varepsilon}{3}\right)d_j^2 \right].$$

Consequently, selecting the $K$ smallest projected noisy distances yields a $(1 + O(\varepsilon))$-approximate $K$-nearest-neighbor set for the latent instance, and the order of the $K$ closest neighbors is preserved up to a $(1 + O(\varepsilon))$ distortion.

## 3 LOWER BOUNDS

In Theorem 1.1, we showed an algorithmic result for which $\sigma \in O(1/k^{1/4})$ (resp. $\sigma \in O(\varepsilon)$). In this section, we demonstrate that this dependence on $k$ (resp. $\varepsilon$) for $\sigma$ is optimal by establishing an information-theoretic lower bound showing that $\sigma = O(1/k^{1/4})$ (resp. $\sigma \in O(\varepsilon)$) is necessary. We prove this hardness result for a computationally easier problem than the one we have addressed. Let $p_1, \ldots, p_n, q$ be points in $\mathbb{R}^k$. We note that reducing the dimension of the ambient space only makes the problem easier in our reduction. Let $\delta_1, \ldots, \delta_n, \delta_q$ be noise vectors where each component is drawn independently and identically from $N(0, \sigma^2)$. We observe the perturbed points: $\tilde{p}_i = p_i + \delta_i$, and $\tilde{q} = q + \delta_q$.

### 3.1 DEPENDENCE ON THE SUBSPACE DIMENSION $k$

The following sequence represents a reduction from the original problem to our target problem. Specifically, we are starting from our NN recovery problem and making the problem instance easier

for any potential algorithm to solve. If we can prove that even this simplified, easier problem is impossible to solve reliably, it immediately implies that the original, harder problem (with the larger gap) must also be impossible to solve. Without loss of generality, assume the distance from $q$ to its nearest neighbor is 1. We assume $\varepsilon > 0$ is a fixed constant throughout this chain of reductions and each step introduces at most a constant change in parameters.

- Given $\tilde{p}_1, \ldots, \tilde{p}_n, \tilde{q}$, there is a unique index $j^*$ such that $||q - p_{j^*}|| \leq 1$. All other points satisfy $||q - p_j|| \geq 1 + \varepsilon$. The goal is to output $j^*$.

- Given $\tilde{p}_1, \tilde{p}_2$, the goal is to distinguish between the two cases: (i) $p_1 = \mathbf{0}$ and $||p_2|| \geq 1$, or (ii) $||p_1|| \geq 1$ and $p_2 = \mathbf{0}$.

- Given $\tilde{p}_1, \tilde{p}_2$, the goal is to distinguish between the two cases: (i) $p_1 = \mathbf{0}$ and $p_2 \sim N(0, \frac{1}{k}I_k)$, or (ii) $p_1 \sim N(0, \frac{1}{k}I_k)$ and $p_2 = \mathbf{0}$.

To show the reduction from the first problem to the second, we show that the second problem is a special case of the first: Consider the case where the first problem has $n = 2$, $q = \mathbf{0}$. If $p_1 = \mathbf{0}, ||p_2|| \geq 1$, the only correct answer to NN is $p_1$ ($p_2$ is not a valid answer). If $||p_1|| \geq 1, p_2 = \mathbf{0}$, the only correct answer is 2. The final reduction step is based on the concentration of the chi-squared distribution. For asymptotically large $k$, this concentration implies that any $p \sim N(0, \frac{1}{k}I_k)$ has a norm that is almost 1 except with exponentially small probability.

The following lemma is a hardness result for the last problem:

**Lemma 3.1.** *Suppose $X, Y$ are random vectors in $\mathbb{R}^k$, where $X \sim N(0, \sigma^2 I_k)$ and $Y \sim N(0, (\sigma^2 + \frac{1}{k})I_k)$. If $\sigma \in \omega(1/k^{1/4})$, then given the unordered pair $\{X, Y\}$, no test can tell which distribution the sample came from with high probability.*

## 3.2 DEPENDENCE ON THE DISTANCE GAP $\varepsilon$

In a similar way as in Section 3.1, we reduce the original NN recovery problem to a computationally simpler one to facilitate hardness analysis. The problem reduction details are available in the appendix. We only give the final lemma below:

**Lemma 3.2.** *Suppose $X, Y$ are random vectors in $\mathbb{R}^k$, where $X \sim N(\mathbf{e}_1, \sigma^2 I_k)$ and $Y \sim N((1 + \varepsilon)\mathbf{e}_1, \sigma^2 I_k)$. If $\sigma \in \omega(\varepsilon)$, then given the unordered pair $\{X, Y\}$, no test can tell which distribution the sample came from with high probability.*

## 4 EXPERIMENTS

In this section, we implement and evaluate our main algorithm. First, we show that it empirically outperforms the naïve algorithm, suggesting that the denoising effect of the SVD is significant. Second, we demonstrate that our analysis in Corollary 2.8 is tight with respect to the parameter $s_k(B)$.

## 4.1 EXPERIMENTAL EVALUATION

We first describe the details of our initial experiment. Our main algorithm is simple to implement. As a baseline, we compare it against a naïve algorithm that selects the index minimizing $||Ax_j||$ over $1 \leq j \leq n$, where $x'_j = \mathbf{e}_{n+1} - \mathbf{e}_j$. This naïve method is affected by the ambient space $\mathbb{R}^d$ when determining the noise threshold $\sigma$ required for successful recovery, while our main algorithm depends only on the latent subspace dimension $k$ (specifically, $\sigma = O(k^{-1/4})$). Note that this naïve method does not mean the algorithm in the (Abdullah et al., 2014). We could not use them as a baseline because their time and space complexity made implementation infeasible for our experimental setup.

We evaluate our algorithm on two real datasets from different domains: one image-based and one text-based. Throughout the experiments, we fix the parameters as follows: $n = 3\,000$, $k = 30$, and $\varepsilon = 0.05$. For a given dataset and fixed noise level $\sigma$, we perform 100 queries to compute the success probability. As our algorithm is efficient and simple to implement, all experiments were conducted on a CPU with an M1 chip and 16 GB of RAM. We give the full details of our datasets.

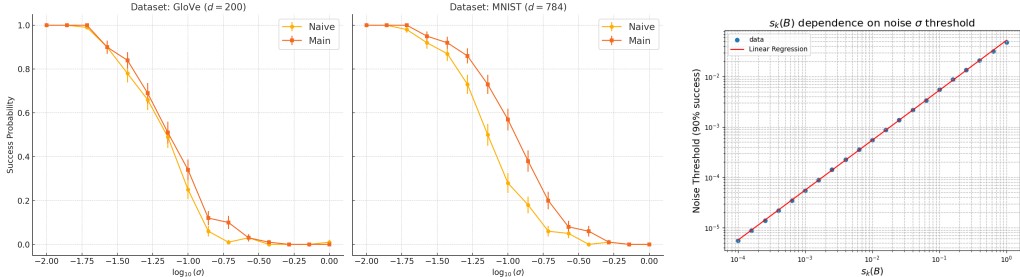

Figure 1: Performance comparison on two real-world datasets (left) and analysis of the noise threshold's dependence on the singular value $s_k(B)$ (right).

**1) GloVe[1]:** This is a set of pre-trained word embeddings where each English word is represented by a high-dimensional real vector. We use the GloVe Twitter 27B dataset, which provides $1\,193\,514$ word vectors of dimension 200. We randomly sample $n = 3\,000$ to construct the data matrix $B$.

**2) MNIST[2]:** MNIST consists of $70\,000$ images of handwritten digits (0-9) in grayscale, of which $60\,000$ are training and $10\,000$ testing. Each image is $28 \times 28$, yielding a 784-dimensional vector of pixel intensities. We sample $n = 3\,000$ training images (flattened to $d = 784$) as data points.

*Preprocessing.* To align the real-world datasets with our theoretical model, we first performed a rank-$k$ approximation on the data matrix $B$. We then selected queries and rescaled the data to ensure the nearest neighbor was at distance 1 and all other points were at least $1 + \varepsilon$ away. The full preprocessing details are available in the appendix.

*Results.* Across both datasets, our main algorithm consistently outperforms the naïve algorithm across the full range of $\sigma$ values. In particular, the failure threshold—the smallest $\sigma$ at which NN recovery becomes unreliable—is significantly higher for our algorithm. We observe two characteristic noise regimes: one where the noise is too small to affect the NN, and another where recovery is information-theoretically impossible. Between these extremes lies a meaningful intermediate regime in which the performance of the two algorithms diverges. In this regime, the performance gap is more pronounced when the ambient dimension $d$ is large. The qualitative results are consistent across both image (MNIST) and text (GloVe) domains, indicating cross-domain robustness. These results illustrate the practical benefits of our approach.

## 4.2 DEPENDENCE ON $s_k(B)$

We now describe the details of our second experiment. While the analyses in Sections 2 and 3 characterize the algorithm's performance in terms of the subspace dimension $k$ and the distance gap $\varepsilon$, it is also of interest to examine its dependence on $s_k(B)$. In Corollary 2.8, we showed that our algorithm exhibits a linear dependence: $\sigma = O(s_k(B))$. Our empirical results confirm these dependencies, suggesting that the analysis is tight.

*Data Generation.* For the second experiment, we generated synthetic data to analyze the algorithm's dependence on $s_k(B)$. We constructed the data matrix $B$ via an SVD-based approach, allowing us to explicitly control its singular values. A subsequent procedure was used to embed a query and its nearest neighbor to satisfy the $1 + \varepsilon$ distance gap condition. A detailed description of the data generation process can be found in the appendix.

*Results.* In this plot, we observe a clear linear dependence on each parameter across the full range of $\sigma$. The *noise threshold* is defined as the value of $\sigma$ at which the success probability first drops below $90\%$ for a given parameter setting. For each parameter configuration, we repeat the experiment 100 times to estimate the success probability. These results demonstrate that our theoretical analysis is tight. Note that this does not imply the optimality of our algorithm with respect to $s_k(B)$.

---

[1]https://nlp.stanford.edu/projects/glove/
[2]http://yann.lecun.com/exdb/mnist/

ACKNOWLEDGMENTS

The authors were supported in part Office of Naval Research award number N000142112647, and a Simons Investigator Award.

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

## A  SUPPLEMENTARY MATERIAL

### A.1  POSTPONED PROOFS

#### A.1.1  PROOF OF LEMMA 2.4

*Proof.* $B^{(1)}$'s columns all lie in $V$. So, $s_{k+1}(B^{(1)}) = 0$. Thus, instantiating Lemma 2.2 gives:

$$||P_{U^{(2)}} - P_V|| \leq \frac{2 \left|\left|C^{(2)}\right|\right|}{s_k(B^{(2)})}. \tag{3}$$

Note that $C^{(2)}$ is a $d \times (\frac{n}{2} + 1)$ matrix. Applying Lemma 2.3 with $t = \sigma \sqrt{2 \ln(8n^2)}$ gives:

$$||P_{U^{(2)}} - P_V|| \leq \frac{2 \left|\left|C^{(2)}\right|\right|}{s_k(B^{(2)})} \leq \frac{2\sigma}{s_k(B^{(2)})} \left( \sqrt{d} + \sqrt{\frac{n}{2} + 1} + \sqrt{2 \ln(8n^2)} \right) \leq \frac{8\sigma \sqrt{n}}{s_k(B^{(2)})} \tag{4}$$

with at least $1 - \frac{1}{4n^2}$ probability (using $n \geq d, 10$). In conclusion, the lemma is proved as follows:

$$\Pr\left[ \left|\left|(P_{U^{(2)}} - P_V)B^{(1)}x_j\right|\right|^2 \leq \frac{100\sigma^2 n}{s_k^2(B^{(2)})} \left|\left|B^{(1)}x_j\right|\right|^2 \right] \geq 1 - \frac{1}{4n^2}$$

for all $1 \leq j \leq n/2$. $\qquad\square$

#### A.1.2  PROOF OF LEMMA 2.5

*Proof.* From the definition of $C^{(1)}$ and $x_j$, the vector $y := C^{(1)}x_j \in \mathbb{R}^d$ is just a summation of two Gaussian random vectors. Thus, $y \sim N(0, 2\sigma^2 I_d)$. Also, for any basis $\{u_1, \ldots, u_k\}$ of $\text{col}(U^{(2)})$, since $P_{U^{(2)}}$ is a projection matrix, the following holds:

$$\left|\left|P_{U^{(2)}}C^{(1)}x_j\right|\right|^2 = \sum_{i=1}^{k} \left(u_i^T y\right)^2.$$

Note that $P_{U^{(2)}}$ and $C^{(1)}$ are stochastically independent (since $P_{shU}$ depends only on $C^{(2)}$ and not on $C^{(1)}$). Since $y \sim N(0, 2\sigma^2 I_d)$ is spherically symmetric, its entries $X_i = u_i^T y$ are independent $N(0, 2\sigma^2)$ random variables. Hence

$$\sum_{i=1}^{k} (u_i^T y)^2 = ||X||^2 = 2\sigma^2 \cdot \sum_{i=1}^{k} \left( \frac{X_i}{\sqrt{2\sigma^2}} \right)^2 = 2\sigma^2 Z$$

where $Z \sim \chi_k^2$. Now we can apply the Laurent–Massart tail bound (Laurent and Massart, 2000) for $\chi^2$ distribution.

$$\forall x \geq 0, \Pr\left[ |Z - k| > 2\sqrt{kx} + 2x \right] \leq e^{-x}.$$

Setting $x = 6 \ln n$, using the Laurant-Massart bound and taking the union over $j = 1, 2, 3 \ldots n/2$ gives us the Lemma. $\qquad\square$

### A.1.3   PROOF OF LEMMA 2.6

*Proof.* We have

$$\left\| B^{(1)T} x_j^T (P_{U^{(2)}} - P_V) B^{(1)} x_j \right\| \leq \| P_{U^{(2)}} - P_V \| \left\| B^{(1)} x_j \right\|^2. \tag{5}$$

Applying (4) to the right hand side, the proof completes. □

### A.1.4   PROOF OF LEMMA 2.7

*Proof.* Let $y := x_j^T B^{(1)T} P_{U^{(2)}}$. $y$ and $C^{(1)} x_j$ are stochastically independent (Recall that $B$ is not random, but, a fixed matrix.) So, $y C^{(1)} x_j$ is a real-valued Gaussian random variable ith distribution $N(0, 2 \|y\|^2 \sigma^2)$. Further, $\|y\| \leq \left\| B^{(1)} x_j \right\|$ since $P_{U^{(2)}}$ is a projection matrix. Now, the Lemma follows by standard Gaussian tail bounds.

□

### A.1.5   PROOF OF LEMMA 3.1

*Proof.* We want to calculate the KL divergence between two distributions. Let $P_1 = N(0, \sigma^2 I_k)$ and $P_2 = N(0, (\sigma^2 + \frac{1}{k}) I_k)$. We evaluate this by computing the KL divergence between the joint distributions $R_1(x, y) = P_1(x) P_2(y)$ and $R_2(x, y) = P_2(x) P_1(y)$. Note that these two distributions differ only by a swap. Since the KL divergence is additive for independent distributions, the following holds:

$$D_{KL}(R_1 \| R_2) = D_{KL}(P_1 \| P_2) + D_{KL}(P_2 \| P_1).$$

The formula for the KL divergence between two multivariate $k$-dimensional gaussian distributions $Q_1 = N(\mu_1, \Sigma_1)$ and $Q_2 = N(\mu_2, \Sigma_2)$ is:

$$D_{KL}(Q_1 \| Q_2) = \frac{1}{2} \left[ \log \frac{|\Sigma_2|}{|\Sigma_1|} - k + \text{tr}\{\Sigma_2^{-1} \Sigma_1\} + (\mu_2 - \mu_1)^T \Sigma_2^{-1} (\mu_2 - \mu_1) \right].$$

Using the above formula, we get

$$2 D_{KL}(R_1 \| R_2) = k \log \frac{\sigma^2 + (1/k)}{\sigma^2} - k + k \frac{\sigma^2}{\sigma^2 + (1/k)}$$

$$+ k \log \frac{\sigma^2}{\sigma^2 + (1/k)} - k + k \left( 1 + (1/k\sigma^2) \right)$$

$$= -2k + k \frac{1}{1 + (1/k\sigma^2)} + k + \frac{1}{\sigma^2} \leq \frac{1}{k\sigma^4},$$

using $1/(1 + \eta) \leq 1 - \eta + \eta^2$ for $\eta \in [0, 1]$. Therefore, the KL divergence approaches 0 when $\sigma = \omega(k^{-1/4})$. This means that it becomes impossible to determine which distribution the observed vector came from as $k$ grows large. □

### A.1.6   PROOF OF LEMMA 3.2

*Proof.* Translating both distributions by $-\mathbf{e}_1$ and scaling by $1/\sigma$ reduces the problem to distinguishing $N(0, I_k)$ from $N((\varepsilon/\sigma)\mathbf{e}_1, I_k)$ while preserving distinguishability. Because the coordinates are independent and identical except for the mean shift in the first coordinate, the optimal test depends only on the first coordinate. Thus, the task is equivalent to distinguishing the 1–dimensional Gaussian distribution $N(0, 1^2)$ and $N(\varepsilon/\sigma, 1^2)$. According to (Devroye et al., 2023), the total variation distance between two one-dimensional Gaussian distributions with unit variance is at most half the difference of their means. Therefore, $d_{TV}\left(N(0, 1^2), N(\frac{\varepsilon}{\sigma}, 1^2)\right) \leq \frac{\varepsilon}{2\sigma}$, which implies that based on the observed vector, it becomes impossible to determine which distribution it came from as $\frac{\varepsilon}{\sigma} \to 0$. □

| Noise Regime | Magnitude | Random Projection | Abdullah et al. (2014) | SVD (Ours) |
|---|---|---|---|---|
| $\sigma \ll 1/\sqrt{d}$ | $o(1)$ | ✓Succeeds | ✓Succeeds | ✓Succeeds |
| $\sigma \in O(1/d^{1/4})$ | $\approx \sigma\sqrt{d}$ | **✗ Fails** | ✓Succeeds | ✓Succeeds |
| $\sigma \in O(1/k^{1/4})$ | $\approx \sigma\sqrt{d}$ | ✗ Fails | **✗ Fails** | ✓Succeeds |
| $\sigma \gg 1/k^{1/4}$ | $\approx \sigma\sqrt{d}$ | ✗ Fails | ✗ Fails | **✗ Fails** (info-theoretic) |

Table 1: Comparison of Noise Tolerance Regimes. SVD succeeds in the "Intermediate" regime where the noise norm is large enough to break Random Projections (RP), but structured enough to be filtered by SVD.

## A.2 POSTPONED DETAILS

### A.2.1 COMPARISON OF NOISE TOLERANCE REGIMES

### A.2.2 PROBLEM REDUCTION DETAILS FOR SECTION 3.2

The following describes the original problem and the simplified target problem:

- Given $\tilde{p}_1, \ldots, \tilde{p}_n, \tilde{q}$, there is a unique index $j*$ such that $||q - p_{j*}|| \leq 1$. All other points satisfy $||q - p_j|| \geq 1 + \epsilon$. The goal is to output $j^*$.

- ($n = 2$ and $q = \mathbf{0}$) Given $\tilde{p}_1, \tilde{p}_2$, there is a unique index $j*$ such that $||p_{j*}|| \leq 1$. The other point satisfies $||p_j|| \geq 1 + \epsilon$. The goal is to output $j*$.

- (Fix the data generation process) Given $\tilde{p}_1, \tilde{p}_2$, distinguish between the two cases: (i) $p_1 = \mathbf{e}_1$ and $p_2 = (1 + \epsilon)\mathbf{e}_1$, or (ii) $p_1 = (1 + \epsilon)\mathbf{e}_1$ and $p_2 = \mathbf{e}_1$.

### A.2.3 PREPROCESSING DETAILS FOR SECTION 4.1

Our theoretical guarantees assume that the data lies entirely in a $k$-dimensional subspace, which does not strictly hold in real datasets. However, both datasets exhibit low intrinsic dimensionality, as indicated by the decay of their singular values. To align with the theoretical assumptions and highlight performance within a low-rank subspace, we apply a rank-$k$ approximation to the sampled matrix $B$ by retaining only its top $k$ singular components. This transformation preserves the essential structure of the data while making the setup consistent with our analysis.

For each dataset, we generate 100 *query points* by projecting held-out data onto the rank-$k$ subspace and selecting points whose nearest-to-second-nearest neighbor distance ratio is just above $1+\varepsilon$. Due to the dataset structure, all GloVe vectors are eligible as query candidates, while for MNIST, queries are sampled from the test set. After that, each column is rescaled such that the NN lies exactly at distance 1 from the query point, and i.i.d. noise $N(0, \sigma^2)$ is added. Finally, we randomly shuffle the first $n$ columns to ensure that both $B^{(1)}$ and $B^{(2)}$ have rank $k$, as required by Theorem 1.1.

### A.2.4 DATA GENERATION DETAILS FOR SECTION 4.2

Unlike in the first experiment, this experiment is conducted on randomly generated datasets. The purpose here is to find concrete examples that clearly reveal the linear dependence of our algorithm on $s_k(B)$. We fix the parameters as follows: $n = 200$, $d = 100$, $k = 10$, and $\varepsilon = 0.05$. To study the dependence on $s_k(B)$, we start with the SVD decomposition $B = X\Sigma Y^\top$, where $X$ and $Y$ are orthogonal matrices randomly generated via QR decomposition of random Gaussian matrices. The singular values of $B$ are controlled by setting the diagonal entries of $\Sigma$.

However, the resulting matrix $B = X\Sigma Y^\top$ may not automatically satisfy the distance gap condition—that is, the requirement that the NN lies at distance exactly 1 from the query point, and the second NN is at distance at least $1 + \varepsilon$. To enforce this condition, we proceed as follows. We first generate the first $n - 1$ columns of $B$ using the above procedure.

Then, we sample a random direction $\vec{u}$ that lies in the intrinsic $k$-dimensional subspace $V$. Essentially, we embed the query point $q = t_1\vec{u}$ and the $n$-th point $p_n = t_2\vec{u}$ in the space for some scalars

$t_1$ and $t_2$. The value $t_1$ is set by starting from $+\infty$ and decreasing it until the distance between the nearest neighbor among the other $n-1$ points and $t_1\vec{u}$ becomes $1+\epsilon$. Then, we set $t_2 = t_1 + 1/||\vec{u}||$, which makes the distance between the query point and this $n$-th point exactly $1$. This construction makes the query point, its nearest neighbor, and the origin collinear. While one could add noise to these points to avoid this artificial colinearity, we believe it would not affect the performance of the algorithms in our context.

