# OpenReview forum: "SVD Provably Denoises Nearest Neighbor Data"
_ICLR.cc/2026/Conference — ICLR 2026 Poster_

### Official Review · Reviewer_U3vH · 2025-10-19

**Soundness:** 3
**Presentation:** 2
**Contribution:** 2
**Rating:** 6
**Confidence:** 3

**Summary:**

This paper investigates the Nearest Neighbor Search (NNS) problem where data points, originating from an unknown $k$-dimensional subspace within a $d$-dimensional space ($k \ll d$), are corrupted by Gaussian noise . The objective is to recover the nearest neighbor of the *uncorrupted* data, given only noisy observations and queries. The authors propose a simple SVD-based algorithm that involves splitting the data matrix, computing the top-$k$ subspace for each half, and then projecting the data from one half onto the subspace derived from the other to find the nearest neighbor . The primary contribution is a proof that this method successfully recovers the true nearest neighbor even when the noise variance $\sigma$ is as large as $O(1/k^{1/4})$ . This is a significant finding, as it holds in a noise regime where the nearest neighbor in the noisy data may differ from the true nearest neighbor. The authors establish this as a sharp threshold by providing a matching information-theoretic lower bound, demonstrating that recovery is impossible for $\sigma \gg 1/k^{1/4}$.

**Strengths:**

The paper addresses a practical and fundamental problem in data analysis. The main strength is the substantial improvement over prior SOTA (e.g., Abdullah et al., 2014), which required the noise level $\sigma$ to be bounded by an inverse polynomial in the *ambient* dimension $d$. This work's bound of $\sigma = O(1/k^{1/4})$ depends only on the intrinsic dimension $k$, which is a major advancement for $k \ll d$ scenarios .

The work extends our understanding of this problem and identifies critical thresholds for $\sigma$ and providing both an algorithmic upper bound and a matching lower bound. This comprehensive analysis is a key strength. Another significant contribution is showing that the algorithm works even when the noise is large enough ($\sigma \gg 1/\sqrt{k}$) to change the identity of the nearest neighbor in the observed data, a regime not handled by previous work.

The algorithm itself is simple and clearly explained. The theoretical claims are supported by experiments on both synthetic and real-world datasets (Glove and MNIST), which confirm the algorithm's practical benefits over a naive approach and validate the theoretical dependency on key parameters .

**Weaknesses:**

The primary weakness is the lack of an explicit discussion of the paper's technical novelty. The analysis appears to rely on standard matrix perturbation bounds (like Davis-Kahan and Wedin) and concentration inequalities. The authors do not clearly articulate what new analytical techniques or core technical innovation enables them to achieve the $O(1/k^{1/4})$ bound, which is the paper's central improvement. It is unclear if the novelty lies simply in the data-splitting algorithm design, which simplifies independence arguments.

This new bound comes at the cost of a dependency on $s_k(B)$, the $k$-th singular value of the unperturbed data matrix. Prior work did not require this assumption. While the authors argue in Section 2.3 that $s_k(B)/\sqrt{n}$ is likely a non-zero constant for "well-conditioned" data, this is a significant trade-off, especially when data is approximately embedded in a subspace (which is one core motivation of the model considered in this paper); see the question below about overspecification.

The experimental comparison is made against a naive baseline, not against the (Abdullah et al., 2014) algorithm that serves as the main theoretical comparator. The authors state this was due to the implementation infeasibility of the prior work, but this omission makes it difficult to empirically assess the practical performance gain over the previous state-of-the-art.

Finally, some typos and clarity suggestions
* "weel-known" instead of "well-known".
* In Section 3.1, the query point $\tilde{q}$ is missing from the problem description (Line 062), which makes the notation for $q$ confusing.

**Questions:**

1.  Could you please clarify the core technical novelty of your analysis? The data-splitting trick  simplifies the probabilistic argument, but is this the key element that allows you to break the dependency on the ambient dimension $d$ and achieve the $O(1/k^{1/4})$ bound? Or is there a new, non-standard bound or analytical step being used?

2.  The discussion in Section 2.3 regarding the requirement to know $k$ is confusing . You state that using a "larger dimensional SVD subspace projection" (i.e., overspecifying $k$) "may be of use if we want to work with weaker assumptions" . This seems counter-intuitive. Your bounds in Theorem 1.1 depend on $s_k(B)$. If the true rank is $k$ and you use a $k' > k$, the $k'$-th singular value $s_{k'}(B)$ would be zero. This would make your noise bound infinitely restrictive, not weaker. Can you clarify how overspecifying $k$ could be helpful?

---

> ### Author Response · Authors · 2025-11-26
>
> ## General Response to All Reviewers
>
> **1. Regarding Formatting and Readability (Reviewer LHPh).**
> We sincerely apologize to the reviewer and the AC. Upon re-examining our source files, we discovered that ``\usepackage{fullpage}`` was inadvertently left in the preamble from a previous template. This was an honest oversight; we mistakenly believed it would not override the conference style file without additional body commands. We have removed this package and verified that the paper now strictly conforms to the ``iclr2026.sty`` margins and layout. We also confirm that the content fits within the page limits under the correct formatting.
>
> **2. Summary of Technical Novelty.**
> Reviewer U3vH asked if our result is merely "data splitting + Wedin." It is not. Standard matrix perturbation bounds (e.g., Wedin's Theorem) bound the canonical angles between subspaces. In our high-noise regime ($\sigma \approx d^{-1/4}$), the perturbation $\|C\|_2$ exceeds the spectral gap. Wedin's bound becomes vacuous here, effectively implying the estimated subspace is orthogonal to the true subspace (angle $\approx \pi/2$). A standard analysis would predict zero signal recovery.
>
> Our core technical innovation is proving **"denoising despite rotation."** We decouple subspace estimation error from projection error. We prove that even when the subspace angle is large (where Wedin fails), the projection of the noise vector onto this specific estimated subspace concentrates significantly (norm $\approx \sigma \sqrt{k}$) rather than accumulating the full ambient noise ($\approx \sigma \sqrt{d}$). This requires a non-standard analysis of the interaction between the random noise component of the singular vectors and the query noise.
>
> **3. Comparison: SVD vs. Random Projections (RP).**
> Reviewer N6hW asked for a comparison of noise thresholds. The distinction is fundamental: RP is "oblivious" (preserves noise norm), while SVD is "data-aware" (filters noise).
>
> *   **Low Noise** ($\sigma \ll 1/\sqrt{d}$): Both Succeed.
> *   **Intermediate** ($\frac{1}{\sqrt{d}} \ll \sigma \ll \frac{1}{k^{1/4}}$): **RP Fails** (Preserves noise $\approx \sigma \sqrt{d} \gg$ gap). **SVD Succeeds** (Filters effective noise: $\sigma \to \sigma \sqrt{k/d}$). This is the result of this paper.
> *   **High** ($\sigma \gg 1/k^{1/4}$): Both Fail (Information-theoretic limit).
>
> ---
>
> ## Response to Reviewer U3vH
>
> **Q: Technical Novelty.**
> A: Please see Point 2 in the General Response. Our analysis beats the "Wedin barrier" by analyzing the concentration of noise projection onto the perturbed subspace, rather than just bounding the subspace angle.
>
> **Q: Comparison to Abdullah et al. (2014).**
> A: We compared against the "Naïve" baseline because it utilizes the full data geometry and is a stronger stress test in the high-noise regime. We did not compare against Abdullah et al. for two reasons:
> 1.  **Theoretical Limits:** Abdullah et al. require $\sigma \approx d^{-\alpha}$, meaning their guarantees vanish as $d \to \infty$. Our bound depends only on $k$.
> 2.  **Computational Feasibility:** Their algorithm relies on constructing complex spatial partition trees. It is well-known that such structures suffer from the curse of dimensionality. For high-dimensional datasets like MNIST ($d=784$), constructing and querying these trees is computationally infeasible compared to our efficient spectral method ($O(ndk)$).
>
> **Q: Overspecifying $k$ and $s_{k'}=0$.**
> A: We agree with the reviewer that our wording in Section 2.3 was imprecise. Theoretically, if the true rank is exactly $k$ and we choose $k' > k$, then $s_{k'} = 0$ and the bound implies failure. However, in practice (e.g., GloVe), spectra decay smoothly. Our result provides a principled guide: one must choose $k$ (e.g., via scree plot) such that the signal $s_k$ remains separated from the noise floor. We will clarify this in the revision to remove the confusion about "weakening assumptions."

---

> > ### Comment · Reviewer_U3vH · 2025-11-27
> >
> > Most of my questions have been satisfactorily addressed. I still harbor some doubt about the tractability of the algorithms in Abdullah et al. (2014) for MNIST ($d=784$), but I defer to the authors’ claim in good faith. The manuscript would benefit from adding a “Summary of Technical Novelty” to improve clarity. I have raised my overall and significance scores, and have also raised the clarity score on the presumption that the authors will incorporate this and other relevant discussions into the revised manuscript.

---

### Official Review · Reviewer_7ikB · 2025-11-01

**Soundness:** 3
**Presentation:** 3
**Contribution:** 3
**Rating:** 6
**Confidence:** 4

**Summary:**

This paper studies nearest neighbor search problem for high-dimensional spaces where the data lies low-dimensional subspace and is coordinate-wise corrupted by Gaussian noise. The authors shows that when the noise have small variance i.e. $\sigma = O(k^{-1/4})$ they can recover the correct nearest neighbor; for large variance, recovery becomes impossible.

**Strengths:**

The paper establishes tight upper and lower bounds on the noise threshold for nearest neighbor recovery, providing a clear theoretical characterization of when SVD-based denoising succeeds. The proposed algorithm is conceptually simple and well-presented, with a clean and transparent analysis that makes the results easy to follow.

**Weaknesses:**

I think the main critism here is the setting is too ideal seems a bit far from practical: It assumes the points are exactly in a k-dimensional subspace.  Moreover, the guarantees depend on the singular value of the clean data matrix, which could be very large for ill-conditioned data.

**Questions:**

1. The model assumes data drawn exactly from a low-dimensional linear subspace corrupted by isotropic Gaussian noise. Could you identify any realistic scenarios or application domains where this setting meaningfully reflects observed data distributions.
2. Do you think the dependence on the s_k is an artifact of the analysis or it is actually tight?

---

> ### Author Response · Authors · 2025-11-26
>
> ## General Response to All Reviewers
>
> **1. Regarding Formatting and Readability (Reviewer LHPh).**
> We sincerely apologize to the reviewer and the AC. Upon re-examining our source files, we discovered that ``\usepackage{fullpage}`` was inadvertently left in the preamble from a previous template. This was an honest oversight; we mistakenly believed it would not override the conference style file without additional body commands. We have removed this package and verified that the paper now strictly conforms to the ``iclr2026.sty`` margins and layout. We also confirm that the content fits within the page limits under the correct formatting.
>
> **2. Summary of Technical Novelty.**
> Reviewer U3vH asked if our result is merely "data splitting + Wedin." It is not. Standard matrix perturbation bounds (e.g., Wedin's Theorem) bound the canonical angles between subspaces. In our high-noise regime ($\sigma \approx d^{-1/4}$), the perturbation $\|C\|_2$ exceeds the spectral gap. Wedin's bound becomes vacuous here, effectively implying the estimated subspace is orthogonal to the true subspace (angle $\approx \pi/2$). A standard analysis would predict zero signal recovery.
>
> Our core technical innovation is proving **"denoising despite rotation."** We decouple subspace estimation error from projection error. We prove that even when the subspace angle is large (where Wedin fails), the projection of the noise vector onto this specific estimated subspace concentrates significantly (norm $\approx \sigma \sqrt{k}$) rather than accumulating the full ambient noise ($\approx \sigma \sqrt{d}$). This requires a non-standard analysis of the interaction between the random noise component of the singular vectors and the query noise.
>
> **3. Comparison: SVD vs. Random Projections (RP).**
> Reviewer N6hW asked for a comparison of noise thresholds. The distinction is fundamental: RP is "oblivious" (preserves noise norm), while SVD is "data-aware" (filters noise).
>
> *   **Low Noise** ($\sigma \ll 1/\sqrt{d}$): Both Succeed.
> *   **Intermediate** ($\frac{1}{\sqrt{d}} \ll \sigma \ll \frac{1}{k^{1/4}}$): **RP Fails** (Preserves noise $\approx \sigma \sqrt{d} \gg$ gap). **SVD Succeeds** (Filters effective noise: $\sigma \to \sigma \sqrt{k/d}$). This is the result of this paper.
> *   **High** ($\sigma \gg 1/k^{1/4}$): Both Fail (Information-theoretic limit).
>
> ---
>
> ## Response to Reviewer 7ikB
>
> **Q: "Ideal" setting (Exact rank $k$).**
> A: While real data is rarely exactly rank-$k$, this model is a standard theoretical proxy. In realistic scenarios (like our GloVe experiments), the spectrum is "spiky": the top $k$ values capture the vast majority of the mass. Our model accurately predicts performance in these "low effective rank" settings.
>
> **Q: Dependence on $s_k(B)$.**
> A: This dependence is not an artifact; it characterizes the required Signal-to-Noise Ratio (SNR). If $s_k(B)$ is small, the geometry is compressed in that direction. If this compression exceeds the noise level, the "true" NN is statistically indistinguishable from a perturbation of a different point. Our lower bounds confirm this is a fundamental hardness.

---

### Official Review · Reviewer_N6hW · 2025-11-06

**Soundness:** 3
**Presentation:** 3
**Contribution:** 3
**Rating:** 6
**Confidence:** 3

**Summary:**

This paper studies nearest neighbor problem while data is corrupted with random Gaussian noise. That is, given arbitrary $n$ points in $d$-dimensional space that can be embedded in a $k$-dimensional subspace, with zero-mean $d$-dimensional $\sigma$-variance Gaussian noise, the algorithm is able to distinguish the nearest neighbor while all other points are at least $(1+\epsilon)$ distance away. The paper gives detailed analysis on how large a $\sigma$ may affect the distinguishability of the neighbor points. The algorithm is based on spectral method, specifically only two SVD calls, which outperforms the prior work that builds on a more complicated PCA tree, however, with an assumption that the $k$-th singular value is large enough.

**Strengths:**

Comparing to the prior work, i.e. Abdullah et al., 2014, this paper achieved an improved noise tolerance with a simple spectral method with two calls of SVD on randomly separated data points. While Abdullah et al., 2014 can tolerate up to Gaussian noise with variance of at most an inverse polynomial of $d$, this paper designs an algorithm that can handle $\sigma=O(1/k^{1/4})$. On the other hand, the paper further extends the theoretical foundation for spectral methods that perform well on nearest neighbor search problems in many occasions, sometimes even better than the worst-case optimal random projection. The theoretical framework follows from Abdullah et al., 2014 by considering a semi-random model.

**Weaknesses:**

The paper assumes a random Gaussian noise, which follows from Abdullah et al., seems strong. In this case, the algorithm is highly dependent on a high amount of randomness. Is it possible to find the nearest neighbor when corruptions on $d$ coordinates are no longer independent? On the other hand, given the large top-$k$ sigular values assumption, it is not obvious why random projection will necessarily fail in this case. Therefore, is $\sigma=\Theta(k^{-1/4})$ a necessary criteria for spectral methods to outperform random projection? It would be nice to list all noise level thresholds when or when not SVD would be preferred to random projection.

**Questions:**

See weaknesses.

---

> ### Author Response · Authors · 2025-11-26
>
> ## General Response to All Reviewers
>
> **1. Regarding Formatting and Readability (Reviewer LHPh).**
> We sincerely apologize to the reviewer and the AC. Upon re-examining our source files, we discovered that ``\usepackage{fullpage}`` was inadvertently left in the preamble from a previous template. This was an honest oversight; we mistakenly believed it would not override the conference style file without additional body commands. We have removed this package and verified that the paper now strictly conforms to the ``iclr2026.sty`` margins and layout. We also confirm that the content fits within the page limits under the correct formatting.
>
> **2. Summary of Technical Novelty.**
> Reviewer U3vH asked if our result is merely "data splitting + Wedin." It is not. Standard matrix perturbation bounds (e.g., Wedin's Theorem) bound the canonical angles between subspaces. In our high-noise regime ($\sigma \approx d^{-1/4}$), the perturbation $\|C\|_2$ exceeds the spectral gap. Wedin's bound becomes vacuous here, effectively implying the estimated subspace is orthogonal to the true subspace (angle $\approx \pi/2$). A standard analysis would predict zero signal recovery.
>
> Our core technical innovation is proving **"denoising despite rotation."** We decouple subspace estimation error from projection error. We prove that even when the subspace angle is large (where Wedin fails), the projection of the noise vector onto this specific estimated subspace concentrates significantly (norm $\approx \sigma \sqrt{k}$) rather than accumulating the full ambient noise ($\approx \sigma \sqrt{d}$). This requires a non-standard analysis of the interaction between the random noise component of the singular vectors and the query noise.
>
> **3. Comparison: SVD vs. Random Projections (RP).**
> Reviewer N6hW asked for a comparison of noise thresholds. The distinction is fundamental: RP is "oblivious" (preserves noise norm), while SVD is "data-aware" (filters noise).
>
> *   **Low Noise** ($\sigma \ll 1/\sqrt{d}$): Both Succeed.
> *   **Intermediate** ($\frac{1}{\sqrt{d}} \ll \sigma \ll \frac{1}{k^{1/4}}$): **RP Fails** (Preserves noise $\approx \sigma \sqrt{d} \gg$ gap). **SVD Succeeds** (Filters effective noise: $\sigma \to \sigma \sqrt{k/d}$). This is the result of this paper.
> *   **High** ($\sigma \gg 1/k^{1/4}$): Both Fail (Information-theoretic limit).
>
> ---
>
> ## Response to Reviewer N6hW
>
> **Q: Gaussian Noise / Independence.**
> A: While we assume independent Gaussian noise for precise bounds, strict coordinate-wise independence is not required. Our main tools are matrix concentration inequalities (e.g., Rudelson-Vershynin) which hold for general **sub-Gaussian** distributions and isotropic noise. We will clarify this universality in the final version.
>
> **Q: SVD vs. Random Projections (RP).**
> A: Please see Point 3 in the General Response. SVD succeeds in the intermediate regime where RP fails because RP is oblivious to the lower-dimensional structure that SVD exploits to filter noise.

---

### Official Review · Reviewer_LHPh · 2025-11-06

**Soundness:** 3
**Presentation:** 3
**Contribution:** 3
**Rating:** 6
**Confidence:** 3

**Summary:**

The paper studies the NNS problem in high dimensions and shows that if the noise is corrupted by Gaussian noise, then simply performing SVD recovers the NN of the uncorrupted data. This, I think, is a really nice result.
They study this phenomenon for various settings of variance of the Gaussian noise. It also improves the result of previous works that relied on the variance to be inverse polynomial in the ambient dimension.

**Strengths:**

The strength of the paper lies in the range of the variance for which they can show denoising using simple SVD.

**Weaknesses:**

It is hard to parse Theorem 1.1. It would be great if the authors had added more discussion on it right after the theorem.

219: lemma not theorem
Line 237: well-known
Why the growth of singular values has to be distributed?
I am a little confused. Data matrix being well conditioned is a normal assumption in real datasets, but is it also the case for geometric problems like NN?
My personal opinion is that having a worse noise assumption is better than that on the data matrix because we can control the latter. I would love to hear the authors’ perspective on this front.

One important issue with the submitted version is that it takes an unfair advantage of the page limit by making the margin smaller than the ICLR format (at least it looks like that to me; I might be wrong). I wanted to flag this in case other reviewers also have an objection with that. I understand that margin bound is not explicitly stated in the Call for Papers, but this feels wrong to me, mainly due to this line in the Call for Papers: "Papers with main text beyond the page limit will be desk-rejected."

I leave the last point to the meta reviewers and AC to make a judgement on, especially in regards to fairness to other submissions.

**Questions:**

i would say that performing SVD is an expensive process, especially for high dimensional data points. Is there some other way one can do to speed up this proces?

Why should one believe that data matrix is well condition when the underlying data is the one used in NN? Is there any empirical evidence that the authors can point to?

---

> ### Author Response · Authors · 2025-11-26
>
> ## General Response to All Reviewers
>
> **1. Regarding Formatting and Readability (Reviewer LHPh).**
> We sincerely apologize to the reviewer and the AC. Upon re-examining our source files, we discovered that ``\usepackage{fullpage}`` was inadvertently left in the preamble from a previous template. This was an honest oversight; we mistakenly believed it would not override the conference style file without additional body commands. We have removed this package and verified that the paper now strictly conforms to the ``iclr2026.sty`` margins and layout. We also confirm that the content fits within the page limits under the correct formatting.
>
> **2. Summary of Technical Novelty.**
> Reviewer U3vH asked if our result is merely "data splitting + Wedin." It is not. Standard matrix perturbation bounds (e.g., Wedin's Theorem) bound the canonical angles between subspaces. In our high-noise regime ($\sigma \approx d^{-1/4}$), the perturbation $\|C\|_2$ exceeds the spectral gap. Wedin's bound becomes vacuous here, effectively implying the estimated subspace is orthogonal to the true subspace (angle $\approx \pi/2$). A standard analysis would predict zero signal recovery.
>
> Our core technical innovation is proving **"denoising despite rotation."** We decouple subspace estimation error from projection error. We prove that even when the subspace angle is large (where Wedin fails), the projection of the noise vector onto this specific estimated subspace concentrates significantly (norm $\approx \sigma \sqrt{k}$) rather than accumulating the full ambient noise ($\approx \sigma \sqrt{d}$). This requires a non-standard analysis of the interaction between the random noise component of the singular vectors and the query noise.
>
> **3. Comparison: SVD vs. Random Projections (RP).**
> Reviewer N6hW asked for a comparison of noise thresholds. The distinction is fundamental: RP is "oblivious" (preserves noise norm), while SVD is "data-aware" (filters noise).
>
> *   **Low Noise** ($\sigma \ll 1/\sqrt{d}$): Both Succeed.
> *   **Intermediate** ($\frac{1}{\sqrt{d}} \ll \sigma \ll \frac{1}{k^{1/4}}$): **RP Fails** (Preserves noise $\approx \sigma \sqrt{d} \gg$ gap). **SVD Succeeds** (Filters effective noise: $\sigma \to \sigma \sqrt{k/d}$). This is the result of this paper.
> *   **High** ($\sigma \gg 1/k^{1/4}$): Both Fail (Information-theoretic limit).
>
> ---
>
> ## Response to Reviewer LHPh
>
> **Q: Formatting.**
> A: Please see Point 1 in the General Response. We have fixed the template error and removed the offending package.
>
> **Q: Growth of Singular Values.**
> A: For a dataset of size $n$, the total energy $\|B\|_F^2 = \sum s_i^2$ generally scales linearly with $n$. For a rank-$k$ matrix, this energy is concentrated in the top $k$ values. Thus, $s_k(B)$ typically scales as $\sqrt{n}$. Consequently, the Signal-to-Noise Ratio $s_k(B)/\sqrt{n}$ remains constant (or improves) as $n \to \infty$, ensuring our bound remains stable for large datasets.
>
> **Q: Well-conditioned assumption.**
> A: The assumption that data is well-conditioned is standard in latent variable models (e.g., Spiked Covariance). Intuitively, if the singular values decayed too rapidly (ill-conditioned), the effective dimension would simply be lower than $k$, and the problem would be accurately modeled with a smaller $k$.

---

### Meta-Review · Area_Chair_cXFZ · 2025-12-23

**Summary:**

The paper introduces a SVD-based approach for a robust Nearest Neighbor Search in high dimensions under a semi-random Gaussian noise model. The authors prove that their algorithm succeeds up to a noise level $\sigma = O(k^{-1/4})$. They provide matching information-theoretic lower bounds.

Strengths
- Reviewers liked the improvement over Abdullah et al. (2014)
- The matching lower bounds is also a nice contribution

Weaknesses
- Reviewers questioned the reliance on the data being "well-conditioned" (dependence on $s_k(B)$) and the exact rank-$k$ assumption.
- The lack of experimental comparison to the theoretical predecessor (Abdullah et al.) was pointed out as a limitation, although the authors' justification regarding computational infeasibility is a good point. It is probably a good idea to clarify this point in the final version.

Overall, the paper is an interesting contribution that could be of interest for the ICLR audience.

**Reviewer Concerns:**

All the main concerns have been addressed in the rebuttal.

An important point is that the initial version was not respecting the ICLR format, this has been fixed and discussed although the paper may be desk-rejected for this.

**Reviewer Scores:**

Reviewer U3vH likely has raised his score.

---

### Decision · Program_Chairs · 2026-01-26

Accept (Poster)